# Transcriptomic and Physiological Analysis of the Effects of Exogenous Phloretin and Pterostilbene on Resistance Responses of *Stylosanthes* against Anthracnose

**DOI:** 10.3390/ijms25052701

**Published:** 2024-02-26

**Authors:** Shizi Zhang, Yunfeng Xu, Fang Wang, Liyun Yang, Lijuan Luo, Lingyan Jiang

**Affiliations:** 1Key Laboratory of Sustainable Utilization of Tropical Biological Resources of Hainan Province, School of Tropical Agriculture and Forestry, Hainan University, Haikou 570228, China; sz88047512@163.com (S.Z.); xuyun336@sina.com (Y.X.); 15132360332@163.com (F.W.); yly1058200235@163.com (L.Y.); luoljd@126.com (L.L.); 2School of Breeding and Multiplication (Sanya Institute of Breeding and Multiplication), Hainan University, Sanya 572025, China

**Keywords:** phloretin, pterostilbene, ROS scavenging, chloroplasts, anthracnose, stylo

## Abstract

Anthracnose caused by *Colletotrichum gloeosporioides* is a destructive disease of *Stylosanthes* (stylo). Combination treatment of phloretin and pterostilbene (PP) has been previously shown to effectively inhibit the conidial germination and mycelial growth of *C. gloeosporioides* in vitro. In this study, the effects of PP treatment on the growth of *C. gloeosporioides* in vivo and the biocontrol mechanisms were investigated. We found that exogenous PP treatment could limit the growth of *C. gloeosporioides* and alleviate the damage of anthracnose in stylo. Comparative transcriptome analysis revealed that 565 genes were up-regulated and 239 genes were down-regulated upon PP treatment during the infection by *C. gloeosporioides*. The differentially expressed genes were mainly related to oxidative stress and chloroplast organization. Further physiological analysis revealed that application of PP after *C. gloeosporioides* inoculation significantly reduced the accumulation of O_2_^•−^ level and increased the accumulation of antioxidants (glutathione, ascorbic acid and flavonoids) as well as the enzyme activity of total antioxidant capacity, superoxide dismutase, catalase, glutathione reductase, peroxidase and ascorbate peroxidase. PP also reduced the decline of chlorophyll a + b and increased the content of carotenoid in response to *C. gloeosporioides* infection. These results suggest that PP treatment alleviates anthracnose by improving antioxidant capacity and reducing the damage of chloroplasts, providing insights into the biocontrol mechanisms of PP on the stylo against anthracnose.

## 1. Introduction

Stylo (*Stylosanthes* spp.) is a dominant leguminous pasture crop mainly distributed in tropical and subtropical regions. As stylo has high yield and quality in stems and leaves, it is commonly used for livestock nutrition [1]. Stylo exhibits excellent adaptation to acidic soil and metal stresses, it is also used as a cover crop to suppress weed growth and a pioneer crop to improve infertile soils [2,3]. *Colletotrichum gloeosporioides* is a hemibiotrophic fungal pathogen which causes lesions on leaves, stems, fruits and other parts of woody or herbaceous plant [4,5,6]. Anthracnose caused by *C. gloeosporioides* is one of the most destructive and predominant diseases of stylo worldwide, greatly limiting the yield of stylo [7]. Typical symptoms of stylo anthracnose are spreading irregular lesions in leaves with dark margins and light brown centers, which spread to all tissues during the late stage of the disease [8,9].

Chemical control, such as chemical fungicides (e.g., benzimidazoles, thiabendazole, benomyl and carbendazim) and sterol inhibitors (e.g., imazalil, prochloraz and propiconazole), is currently an effective approach to control the anthracnose [10,11]. However, large-scale and continuous application of fungicides is detrimental to the environment and lead to pathogen resistance [11,12,13]. Therefore, eco-friendly means of disease control are urgently required. Among various alternatives, biological control has become an attractive way to control plant diseases, and the application of plant natural products has played an important role [14]. The efficacy of plant metabolites against a pathogen relies on either direct antimicrobic activity or induction of plant resistance by indirect mechanisms [15,16]. The defense responses of induced resistance include the activation of defense signaling pathways, the increase in antioxidant capacity, the production of resistance-related secondary metabolites, and the improvement of photosynthesis efficiency [15,16,17].

Within plant natural products, phenolic compounds have been reported to be involved in defense responses against fungi by restricting pathogen growth and enhancing plant resistance [18]. Phloretin is a dihydrochalcone flavonoid first isolated from the root bark of an apple (*Malus domestica Borkh.*) tree and found in the various tissues of plants, including young shoots, roots, bark and seeds [19]. Pterostilbene is a stilbene-derived phenolic compound synthesized by several plant families, including *Vitaceae*, *Dipterocarpaceae*, *Gnetaceae*, *Pinaceae*, *Fabaceae*, *Poaceae*, *Leguminoseae*, and *Cyperaceae* [20]. Many pharmacological studies have shown that both phloretin and pterostilbene have antioxidant, anti-inflammatory, antidiabetic and anticarcinogenic effects [19,21]. Moreover, phloretin has been shown to inhibit pathogenicity and virulence factors against *Candida albicans* [22]. Pterostilbene can effectively inhibit the conidial germination and mycelial growth of *Peronophythora litchii*, *Botrytis cinerea, Macrophomina phaseolina, Rhizoctonia solani,* and *Sclerotinia sclerotiorum* [20,23,24]. Exogenous application of pterostilbene significantly reduces the disease incidence of litchi downy blight and gray mold of table grapes [23,24]. Our previous transcriptome and metabolome studies of stylo have shown that the phenylpropanoid metabolic pathway is significantly up-regulated in response to *Colletotrichum* infection [25]. Thirteen phenolic compounds have been selected to examine the inhibitory effects on the conidial germination and mycelial growth of six *Colletotrichum* species in vitro, including stylo *C. gloeosporioides*, rubber *C. gloeosporioides*, rubber *C. acutatum*, yam *C. gloeosporioides*, mango *C. gloeosporioides* and papaya *C. gloeosporioides*. The combination of phloretin and pterostilbene showed the strongest inhibition on conidial germination and mycelial growth of stylo *C. gloeosporioides*, and the inhibitory effect of the combined treatment was stronger than the separate treatment [26]. However, it is unknown whether phloretin and pterostilbene treatment exerts inhibitory effects on *C. gloeosporioides* in vivo.

The main objectives of this study is to examine the effects of phloretin and pterostilbene (PP) treatment on controlling stylo anthracnose and to explore the biocontrol mechanisms. The disease symptom and the growth of *C. gloeosporioides* in stylo leaves under exogenous PP treatment were evaluated. The transcriptomic and physiological analysis were performed to investigate the mechanisms. These results will provide an alternative prevention strategy for stylo anthracnose.

## 2. Result

### 2.1. Effects of Exogenous Phloretin and Pterostilbene Treatment on Stylo Leaves against Anthracnose

To test the effects of phloretin and pterostilbene on the leaves of stylo against anthracnose, we applied the combination treatment of phloretin and pterostilbene (PP) with inoculation of *C. gloeosporioides* (Cg + PP). The inhibition of anthracnose disease was easily discerned in the Cg + PP treatment group as compared with the Cg group (Figure 1A). The percentage of lesion area of the leaves in Cg + PP was 3.6%, whereas that in Cg was 41.2% (Figure 1B). The incidence rate of anthracnose in Cg + PP was 16%, whereas that in Cg was 85% (Figure 1C). To further investigate the effects of PP treatment on the growth of *C. gloeosporioides* in the leaves during infection, microscopic observation and quantitative detection of *C. gloeosporioides* were performed. The microscopic observation showed that the amount of hyphae of *C. gloeosporioides* was less in the Cg + PP group than that in the Cg at 72 hpi (Figure 1D). Consistently, the qRT-PCR results also showed the reduced accumulation of fungal DNA in the Cg + PP-treated samples from 24 hpi to 72 hpi (Figure 1E). Altogether, these results showed that PP treatment could restrict the infection by *C. gloeosporioides* in the leaves of stylo, reducing the damage of anthracnose.

### 2.2. Transcriptome Analysis of Stylo Leaves in Response to Cg and Cg + PP Treatment

To investigate possible molecular responses of stylo induced by phloretin and pterostilbene during *C. gloeosporioides* infection, a comparative transcriptomic analysis of infected leaves from Cg and Cg + PP treatments was performed. Hierarchical cluster analysis of all identified unigenes [reads per kilobase per million (RPKM) > 0)] showed that the three biological repeats of Cg-treated samples were clustered into one branch, while those of Cg + PP-treated samples were clustered into another branch (Figure 2A, Appendix A). The heatmap analysis of the relative gene expression suggested different gene expression patterns between Cg- and Cg + PP-treated samples. A total of 7980 unigenes were identified in Cg group, while 6594 unigenes were identified in Cg + PP group. The Venn diagram analysis showed that 5660 unigenes were shared between the two groups (Figure 2B). A total of 804 differently expressed genes (DEGs) were identified in Cg + PP as compared to Cg, in which 565 genes were up-regulated (log_2_fold change > 1.5, *p* < 0.05) and 239 genes were down-regulated (log_2_fold change < −1.5, *p* < 0.05) (Figure 2C, Appendix A).

The KEGG pathway enrichment analysis revealed that the enriched DEGs belonged to 22 different KEGG pathways (Figure 3A). DEGs were mainly enriched in metabolic pathways and secondary metabolite synthesis pathways, which were mainly related to oxidative stress and chloroplast organization (Figure 3A, Appendix A). Gene ontology (GO) analysis of the DEGs showed that the DEGs were mainly enriched in signaling transduction (GO:0007165), defense response to fungi (GO:0050832), chloroplast organization (GO:0009658), calmodulin binding (GO:0005516), and response to salicylic acid (GO:0009751) (Figure 3B, Appendix A).

### 2.3. DEGs Involved in the Production of Antioxidant and Chloroplast Organization 

Both KEGG and GO analysis suggested that redox homeostasis and chloroplast may play important roles in the molecular responses of stylo caused by PP treatment against *C. gloeosporioides* infection. We further analyzed the expression levels of genes involved in production of antioxidant and chloroplast organization (Figure 4, Appendix A). Two DEGs involved in AsA-GSH cycle, including ascorbate peroxidase (APX) and glutathione reductase (GR), were identified. The expression of *APX1* was up-regulated in Cg + PP-treated group, while the expression of *GR* was down-regulated. Two DEGs related to GSH homeostasis and GSH-dependent redox regulation were identified. One was *CLT*, encoding a CRT-like transporter required for GSH homeostasis, and another was *GRX2*, encoding glutaredoxin, catalyzing GSH-dependent redox regulation via glutathionylation. The expression levels of both *CLT2* and *GRX2* were up-regulated in Cg + PP. Three DEGs involved in flavonoid biosynthesis, including *CHS*, *7-IOMT* and *CYP17D9*, were up-regulated in Cg + PP. A total of six DEGs associated with chloroplast organization were identified. Four genes involved in assembly and stabilization of photosystem I and II, including *psaF, psaG*, *CYP38* and *FKBP20*, were up-regulated in Cg + PP. The gene *CAB215*, which participated in energy transfer in photosystem II, was also up-regulated. The gene *NYC1*, which encodes chlorophyll catabolic enzymes and is involved in chlorophyll b degradation, was down-regulated. In the carotenoid metabolic pathway, one DEG *AAO3*, encoding an abscisic aldehyde oxidase, was identified as an up-regulated gene in Cg + PP. Altogether, these results suggest that PP treatment increases the production of antioxidants and protects the chloroplast organization of stylo in response to *C. gloeosporioides* infection.

### 2.4. Validation of DEG Expression by qRT-PCR

To validate the main results obtained from transcriptomic analysis (Figure 4), fourteen DEGs were selected to verify their relative expression levels by quantitative real-time PCR (qRT-PCR), using the reference genes of stylo (*UBCE1*) as an internal control. The qRT-PCR and transcriptome data of fourteen genes were displayed in Figure 5. The changes in the transcription levels of these DEGs obtained by qRT-PCR analysis were consistent with those from RNA-Seq by transcriptomic analysis, which confirmed that the expression of these genes was regulated as the response of stylo caused by phloretin and pterostilbene during *C. gloeosporioides* infection.

### 2.5. Phloretin and Pterostilbene Enhanced Antioxidant Capacity of Stylo in Response to C. gloeosporioides

Both transcriptome and qPCR analysis suggested that PP treatment may increase the production of antioxidants in stylo in response to *C. gloeosporioides* infection. We first examined the effects of PP treatment on the accumulation of ROS in response to *C. gloeosporioides* infection. The staining of O_2_^•−^ showed that *C. gloeosporioides* inoculation could induce a large amount of O_2_^•−^ accumulation in stylo leaves, and Cg + PP treatment could significantly reduce the content of O_2_^•−^ in the inoculated leaves, indicating the increase in antioxidant capacity induced by PP treatment (Figure 6). Therefore, we evaluated the content of antioxidant substances and the enzyme activities of antioxidant enzymes in leaves treated with PP, Cg and Cg + PP as compared to the uninoculated control (CK). As shown in Figure 7, the contents of GSH, AsA and flavonoids in PP-treated leaves were not significantly different from those in the CK. Compared with CK and PP-treated leaves, the contents of GSH, AsA and flavonoids were increased in Cg-treated and Cg + PP-treated leaves. The GSH, AsA, and flavonoid contents in the Cg + PP-treated leaves were 1.46-fold, 1.95-fold and 1.45-fold higher than in the Cg, respectively. For antioxidant enzyme activities, we examined the activities of the total antioxidant capacity (T-AOC), superoxide dismutase (SOD), catalase (CAT), glutathione reductase (GR) and peroxidase (POD) and ascorbate peroxidase (APX). The results showed that PP treatment had significantly increased T-AOC, GR, APX, and SOD activity compared to the CK, and the activities of POD and CAT were not altered by PP treatment. Both Cg and Cg + PP treatment increased the activities of all these enzymes, and enzyme activities of SOD, GR, APX, POD, T-AOC, CAT in Cg + PP-treated leaves were 1.42-fold, 1.18-fold, 3.30-fold, 1.30-fold, 2.05-fold, and 1.57-fold higher than those in Cg. These physiological results were consistent with the transcriptional results, suggesting that PP treatment could enhance the antioxidant capacity of stylo after inoculation with *C. gloeosporioides*.

### 2.6. Changes in Pigment Content in Leaves under Different Treatment

Many DEGS related to chloroplast organization were identified in the transcriptome; we therefore examined the contents of chlorophyll a + b and carotenoid in leaves under different treatments (Figure 8). After inoculation with *C. gloeosporioides*, the contents of chlorophyll a + b were reduced both in Cg + PP and Cg compared with CK, while PP reduced the decline of chlorophyll a + b content caused by C. *gloeosporioides* infection (Figure 8A). Treatments with PP and Cg only did not significantly affect the content of the carotenoid compared to that in the CK, and the carotenoid content in Cg + PP was 1.37-fold higher than that in the Cg (Figure 8B). These results were consistent with the general up-regulation of expression levels of genes related to chloroplast assembly and stablization in Cg + PP treatments compared to Cg treatment, indicating that PP could alleviate the damage of chlorophyll by *C. gloeosporioides* and increase the content of the antioxidant pigment carotenoid.

## 3. Discussion

Phenolic compounds substances are important secondary metabolites showing antioxidant capacity, and they tend to accumulate in large quantities under stress conditions [18]. Both phloretin and pterostilbene belong among phenolic compounds, and they have excellent antioxidant properties, thus protecting cells from damages by free radicals [19,21]. In addition, phloretin and pterostilbene also have antimicrobial actions against various fungal pathogens [20,22,23,24]. However, it is largely unknown how phloretin and pterostilbene inhibit *C. gloeosporioides* growth and induce the disease resistance against anthracnose in plants. Our previous studies have shown the combination treatment of phloretin and pterostilbene greatly inhibited the conidial germination and mycelial growth of stylo *C. gloeosporioides* in vitro. In this study, we aimed to investigate the impacts of exogenous treatment of phloretin and pterostilbene on mitigating stylo anthracnose and the biocontrol mechanism via RNA-seq and subsequent physiological analysis.

Comparative transcriptome sequencing analysis identified 804 DEGs between Cg + PP and Cg. Among these DEGs, many DEGs involved in enzymatic reactions to produce antioxidant were up-regulated in Cg + PP as compared to Cg. For example, *APX1*, *CLT2*, *GRX2,* which were associated with GSH and AsA metabolism, were all up-regulated. In addition, genes involved in flavonoid biosynthesis were also up-regulated in Cg + PP, including *CHS*, *7-IOMT* and *CYP17D9*. Consistent with transcriptome results, our physiological results also showed that PP treatment increased the accumulation of antioxidant substances (GSH, AsA and flavonoid) and the activities of antioxidant enzymes (T-AOC, SOD, CAT, GR, POD and APX). In addition, the accumulation of ROS was also reduced by PP treatment post inoculation. These results suggest that PP treatment increases the antioxidant capacity in response to *C. gloeosporioides*, thus increasing the resistance against anthracnose. Indeed, maintaining ROS homeostasis is crucial for the resistance of plants against pathogens, especially the hemibiotrophic pathogens as *C. gloeosporioides,* which first establish a brief biotrophic stage and then switch to a destructive necrotrophic stage, because excessive accumulation of ROS could lead to cellular necrosis in the plants, favoring the infection by pathogens which acquire nutrients from dead tissues. Various reports have proved that the improvement of antioxidant capacity could increase the resistance of plants against the infection by hemibiotrophic or necrophic pathogens [27,28,29]. For example, ROS-scavenging enzymes and antioxidants were strongly induced in *Fusarium*-infected banana to prevent unfettered apoptosis [30]. The production of phenolic compounds and the activity of antioxidative enzymes were increased in biocontrol agent-treated chilli in response to the infection by *Colletotrichum truncatum,* resulting in less lesion development and ROS accumulation [31]. 

Chloroplast is indispensable for producing energy through photosynthesis and defense-related signaling molecules such as calcium, salicylic acid (SA) and ROS [32,33]. Pathogens could manipulate chloroplast structure and functions (e.g., thylakoid remodeling, pigment content, expression of photosynthetic genes, photosynthetic water splitting and electron transport, enzyme redox status, and SA biosynthesis) to inhibit host defense and enhance pathogenicity [34]. *Colletotrichum* infection tends to decrease the chlorophyll content, which is consistent with the chlorosis of plants [31,35,36]. In our function analysis of the DEGs, the DEGs were mainly located in chloroplasts, and many genes encoding important components of photosystem I and II were up-regulated in Cg + PP-treated samples as compared to Cg-treated samples, including *psaF*, *psaG*, *CYP38*, *FKBP20* and *CAB215*, while the gene *NYC1*, encoding chlorophyll catabolic enzymes and involved in chlorophyll b degradation, was down-regulated in Cg + PP. In addition, the gene *AAO3*, encoding an abscisic-aldehyde oxidase and involved in the carotenoid metabolic pathway, was up-regulated in Cg + PP. These results suggest that PP may protect chloroplasts from the damage caused by the infection by *C. gloeosporioides*. Consistently, the physiological results also showed that the decrease in chlorophyll content caused by *C. gloeosporioides* could be reduced by PP treatment, and the content of carotenoid was increased by PP treatment. Therefore, the decrease in the damage to chloroplasts during the infection could also contribute to the resistance induced by PP treatment. Moreover, the increase in the carotenoid content could further strengthen the antioxidant capacity of stylo, as the carotenoid could prevent over-accumulation of ROS by quenching singlet oxygen [37]. The correlation of chlorophyll content with the resistance against *Colletotrichum* species has also been reported in other plant–*Colletotrichum* interaction systems. For example, the chlorophyll content was reduced at a higher rate in the susceptible variety JS-335 than in the resistant variety MAUS-71 of soybean post infection by *Colletotrichum truncatum* [36]. The treatment of a biocontrol agent increased in the levels of photosynthetic pigments in black pepper plants, enhancing the resistance against *Colletotrichum siamense* [38]. However, it is still unknown whether chlorophyll content has a linear relationship with disease resistance.

## 4. Materials and Methods

### 4.1. Stylo Material and Colletotrichum Gloeosporioides Inoculation

Twenty-five-day-old stylo plants (genotype ‘2001-71’) were used. The stylo seeds were provided by the Tropical Crop Germplasm Research Institute, Haikou, Hainan province, China. Stylo seeds were soaked in hot water (80 °C) for 3 min and germinated for 3 days. After germination, the seeds were transplanted into seedling boxes filled with equal volumes of vermiculite and nutrient soil. The seedlings were grown in a greenhouse with an average temperature of 28 °C for approximately 3 weeks [9]. *Colletotrichum gloeosporioides* strain WC-02 was used in this study and described previously [39]. The inoculation of *C. gloeosporioides* was performed according to the previous study [9,25]. The fungus was grown on potato dextrose agar for 3 days, and the spore was produced in complete medium for 3 days. The spores were harvested and diluted with sterile water. The concentration of spore suspension was determined using a hemocytometer under a light microscope and adjusted to 10^6^ spores mL^−1^. The stylo plants were spray-inoculated with a spore suspension containing 0.02% silweet L-77 with 500 μmol L^−1^ phloretin (Sigma-Aldrich, St. Louis, MO, USA) and 500 μmol L^−1^ pterostilbene (Sigma-Aldrich, St. Louis, MO, USA) (Cg + PP treatment) or an equal amount of ethanol solvent (Cg treatment) [26]. The inoculated plants were placed in a dark room at 28 °C and 90% humidity for 12 h. Then, the plants were transferred to a growth chamber at 28 °C, 90% humidity and 16 h photoperiod with a light intensity of 800 μmol m^−2^ s^−1^. Stylo plants sprayed with 500 μmol L^−1^ phloretin and 500 μmol L^−1^ pterostilbene containing 0.02% Silweet L-77 (PP treatment) or an equal amount of ethanol solvent (CK) were used as physiological controls.

### 4.2. Microscopic Observation of C. gloeosporioides Infection in Stylo Leaves

Stylo leaves were sampled at 72 h post inoculation (hpi) and fixed with formalin acetic alcohol solution (ethanol/formaldehyde/acetic acid, 18:5:5, *v*/*v*/*v*) for 12 h, then stained in aniline blue solution for 15 min and washed with sterile distilled water. The samples were transferred to a slide and observed with a light microscope.

### 4.3. Detection of the Fungal DNA of C. gloeosporioides Infection in Stylo Leaves

The accumulation of fungal DNA of *C. gloeosporioides* during infection was detected and calculated according to the previous study [40]. Infected leaves were collected at various time points after inoculation and ground with liquid nitrogen. The genomic DNA (gDNA) was extracted from fungal cultures and leaf samples using the CTAB (cetyl trimethylammonium bromide) method [41]. The cycle threshold (Ct values) of stylo and *C. gloeosporioides* were detected by real-time quantitative PCR (Section 2.5) using primers *UBCE1* and *ACTING4*, respectively (Appendix A). The content of fungal DNA in 100 ng plant DNA was calculated as described previously [40].

### 4.4. Transcriptome Analysis

Samples of infected leaves inoculated with *C. gloeosporioides* together with phloretin and pterostilbene treatment (Cg + PP) or inoculated only with *C. gloeosporioides* (Cg) were randomly sampled at 72 hpi, each with three biological replicates. RNA sequencing was performed by Novogene Co., Ltd. (Beijing, China). Total RNA was isolated using the RNA prep Pure Plant Kit (TIANGEN, Beijing, China), and the quality of RNA was examined using the Agilent 2100 RNA Nano 6000 Assay Kit (Agilent Technologies, Palo Alto, CA, USA). The libraries were prepared using a NEB Next Ultra Directional RNA Library Prep kit (NEB, Beverly, MA, USA) and were sequenced using an Illumina Hiseq™ 4000 platform (Illumina, San Diego, CA, USA). Clean reads were obtained by removing reads containing adapter, reads containing N base and low-quality reads. Transcriptome assembly was accomplished using Trinity software (version 2.4.0). Hierarchical clustering was performed with Corset (version 4.6). Each new cluster was defined as “Gene”. Gene functions were annotated based on Nr (NCBI non-redundant protein sequences, diamond v0.8.22), Nt (NCBI non-redundant nucleotide sequences, NCBI blast 2.2.28+), Pfam (Protein family, HMMER 3.0 package), KOG/COG (Clusters of Orthologous Groups of proteins, diamond v0.8.22), Swiss-Prot (a manually annotated and reviewed protein sequence database, diamond v0.8.22), KO (KEGG Ortholog database, KAAS r140224), GO (Gene Ontology, blast2go b2g4pipe_v2.5). The transcript abundance of each gene was calculated by the fragments per kilobase million mapped reads (FPKM) value using RNA-Seq data with expectation maximization (RSEM). A heat map was generated using the R package “pheatmap” (version 1.0.12) to show the clustering of the genes across all samples with different treatments based on the FPKM data. Differentially expressed genes (DEGs) were screened by DESeq (version 1.18.0) with criteria of an absolute value of log2ratio ≥ 1.5 and a *p*-value < 0.05. GO (Gene Ontology) enrichment analysis and KEGG (Kyoto Encyclopedia of Genes and Genomes) pathway enrichment analysis of the DEGs were performed using KOBAS software (version 3.0).

### 4.5. Real-Time Quantitative PCR

Total RNA was isolated using the RNA prep Pure Plant Kit (TIANGEN, Beijing, China). First-strand complementary DNA synthesis was performed using the HiScript III RT SuperMix for qPCR (Vazyme, Nanjing, China). Real-time quantitative PCR (qRT-PCR) was performed using the Applied Biosystems QuantStudio (TM) 7 Flex System (Thermo Fisher, Waltham, MA, USA) with the specific primers (Appendix A). Primers *UBCE1* and *ACTING4* were used to detect the accumulation of *C. gloeosporioides* gDNA in stylo, and the remaining primers were used to validate the transcriptome results with *UBCE1* as an internal control. The qRT-PCR reaction system consisted of 1μL template DNA, 0.2 μL each primer, 5 μL 2 × ChamQ Universal SYBR qPCR Master Mix (Vazyme, Nanjing, China) and 3.6 μL sterile deionized water. The qPCR conditions were as follows: 95 °C for 5 min, followed by 40 cycles at 95 °C for 15 s and 60 °C for 30 s. Gene expression was determined by the 2^−ΔΔCt^ method. Each qRT-PCR experiment was conducted in triplicates.

### 4.6. Histochemical Detection of ROS Accumulation

The ROS accumulation (O_2_^•−^ level) in the leaves was detected using the histochemical method described previously with modification of incubation time and decolorization [9]. The second fully expanded trifoliate leaves under different treatments were collected and stained in the 0.25 mM nitroblue tetrazolium chloride (NBT, Sigma-Aldrich, St. Louis, MO, USA) solution for 12 h at 25 °C. The saturated chloral hydrate (Aladdin, Shanghai, China) solution was used for decolorization at 60 °C for 12 h. The levels of O_2_^•−^ were visualized in the leaves as indicated by the intensity of blue stain. The staining area was measured using Image J (version 1.54).

### 4.7. Determination of Enzyme Activities and Antioxidant Concentrations

The activities of total antioxidant capacity (T-AOC), superoxide dismutase (SOD), catalase (CAT), glutathione reductase (GR), peroxidase (POD) and ascorbate peroxidase (APX) from each treatment group were determined using the corresponding kits (Nanjing Jiancheng Bioengineering Institute, Nanjing, China) following the instruction manual: Total antioxidant capacity assay kit (A015-1-2), Superoxide Dismutase typed assay kit (A001-2-1), Catalase (CAT) assay kit (A007-1-1), Glutathion reductases assay kit (A062-1-1), Peroxidase assay kit (A084-3-1), Ascorbate peroxidase test kit (A123-1-1). The concentrations of GSH, AsA and total flavonoids from each treatment group were determined using the following kits (Nanjing Jiancheng Bioengineering Institute, Nanjing, China): Reduced glutathione (GSH) assay kit (A006-1-1), Vitamin C assay kit (A009-1-1), plant flavonoids test kit (A142-1-1). Protein content was determined using the coomassie brilliant blue method [42].

### 4.8. Estimation of the Content of Chlorophyll and Carotenoid

The content of chlorophyll and carotenoid was estimated using Arnon’s method [43]. The leaf samples of 0.1 g were thoroughly ground in 5 mL of 95% ethanol and placed at room temperature until the color completely faded. The aqueous layer was collected, and the absorbance was recorded using spectrophotometry at 645 nm, 663 nm and 470 nm for chlorophyll a, b and carotenoid, respectively. The content of chlorophyll and carotenoid was calculated using the following formula [44].
Chlorophyll a (mg/g) = 12.7 (OD663) − 2.69 (OD645) × V/1000 × W(1)
Chlorophyll b (mg/g) = 22.9 (OD645) − 4.68 (OD663) × V/1000 × W(2)
Total chlorophyll (mg/g) = Chlorophyll a + Chlorophyll b(3)
Carotenoid (mg/g) = [1000(OD470) − 2.05(OD663) − (OD645)]/245× V/1000 × W(4)
where OD = optical density, W = weight of sample, and V = volume of ethanol (mL).

### 4.9. Statistical Analysis

Statistical analysis was performed using Microsoft Excel (version 16.72) and SPSS (version 26.0) with one-way ANOVA. Figures were generated using GraphPad Prism software (version 9.5.0). All experimental data were obtained from three biological replicates.

## 5. Conclusions

In the present study, we provided evidence that phloretin and pterostilbene could alleviate anthracnose in stylo. The possible mechanisms of the induced responses of stylo under PP treatment were investigated by transcriptomic and physiological analysis post *C. gloeosporioides* infection. Transcriptome results showed that the differentially expressed genes were mainly related to antioxidants and chloroplasts. The results of physiological tests proved that PP treatment enhanced the ROS-scavenging capability of stylo during *C. gloeosporioides* infection by increasing the accumulation of the antioxidant substances GSH, AsA and flavonoids and the activities of the antioxidant enzymes SOD, APX CAT, POD, GR. In addition, PP treatment also reduced the decline of chlorophyll and increased the accumulation of carotenoid during the infection. The present study revealed the possible mechanisms by which phloretin and pterostilbene enhance the disease resistance of stylo against *C. gloeosporioides*, providing a theoretical basis for the development of phloretin and pterostilbene as potential biological control agents for anthracnose in stylo.

## Figures and Tables

**Figure 1 ijms-25-02701-f001:**
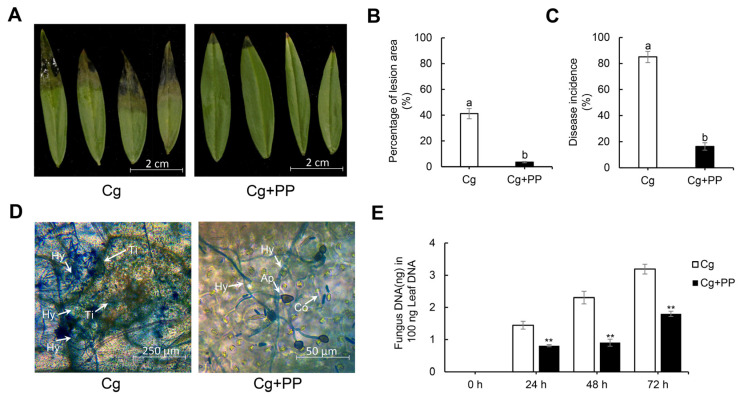
Effects of exogenous phloretin and pterostilbene on stylo leaves against anthracnose. (**A**) Representative images of stylo leaves inoculated with *C. gloeosporioides* (Cg) and *C. gloeosporioides* together with phloretin and pterostilbene (Cg + PP). Images were taken at 72 hpi. (**B**) Statistics of the percentage of lesion area. Leaves were collected at 72 hpi. Data are mean ± SD from three biological replications. Different letters indicate significant difference at the *p* < 0.05 level based on the ANOVA. (**C**) Statistics of the incidence rate of anthracnose. Statistical data were collected at 72 hpi. Data are mean ± SD of 100 samples from three biological replications. Different letters indicate significant difference at the *p* < 0.05 level based on the ANOVA. (**D**) Microscopic observation of the growth of *C. gloeosporioides* in leaves of stylo at 72 hpi. Ap, appressoria; Co, condia; Hy, hyphae; Ti, tissue liquid. (**E**) The accumulation of fungal DNA at various time points post inoculation with *C. gloeosporioides*. Data are mean ± SD of six leaves from three biological replications. The asterisks “**” represent significant difference between Cg and Cg + PP treatments at different time points according to Student’s *t*-test at *p* < 0.01.

**Figure 2 ijms-25-02701-f002:**
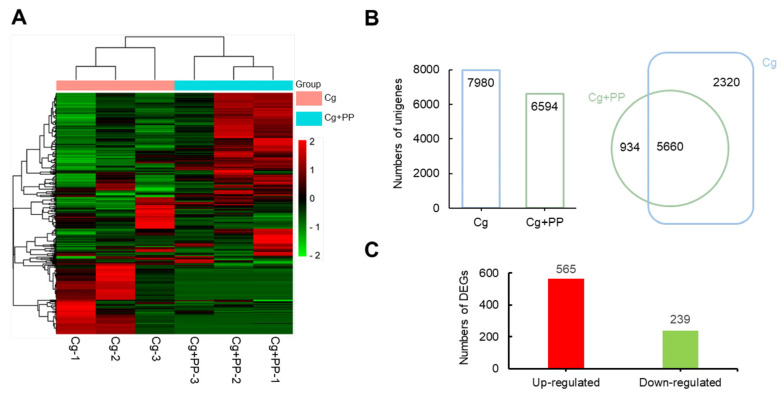
Transcriptional profiles of stylo leaves in response to Cg and Cg + PP treatments. (**A**) Heatmap of the hierarchical cluster analysis of DEGs across all three samples with different treatments. Scale for levels of expression is indicated on the right side of the cluster analysis, ranging from low (green) to high (red), covering an ascending log_10_. (**B**) Statistics of unigenes (RPKM > 0) identified under different treatments. (**C**) Statistics of DEGs in Cg + PP as compared to Cg.

**Figure 3 ijms-25-02701-f003:**
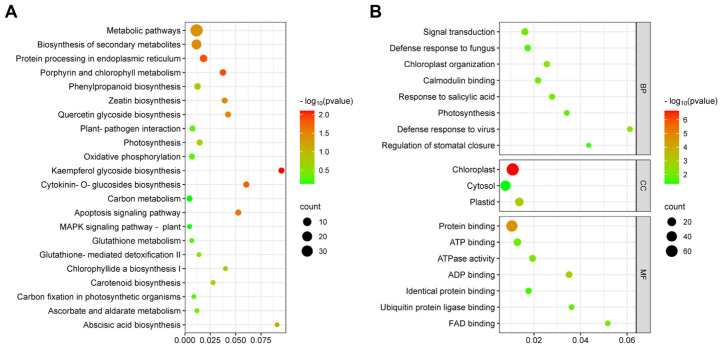
KEGG and GO analysis of DEGs. (**A**) Bubble diagram of KEGG pathways of DEGs. (**B**) GO analysis of DEGs. Y-axis: KEGG/GO terms; X-axis: rich factor indicating the ratio of the number of DEGs relative to the total number of genes annotated in a pathway term. Colors of each bubble ranging from green to red indicate increasing significant (*p* value) enrichment. The bubble size represents the gene number.

**Figure 4 ijms-25-02701-f004:**
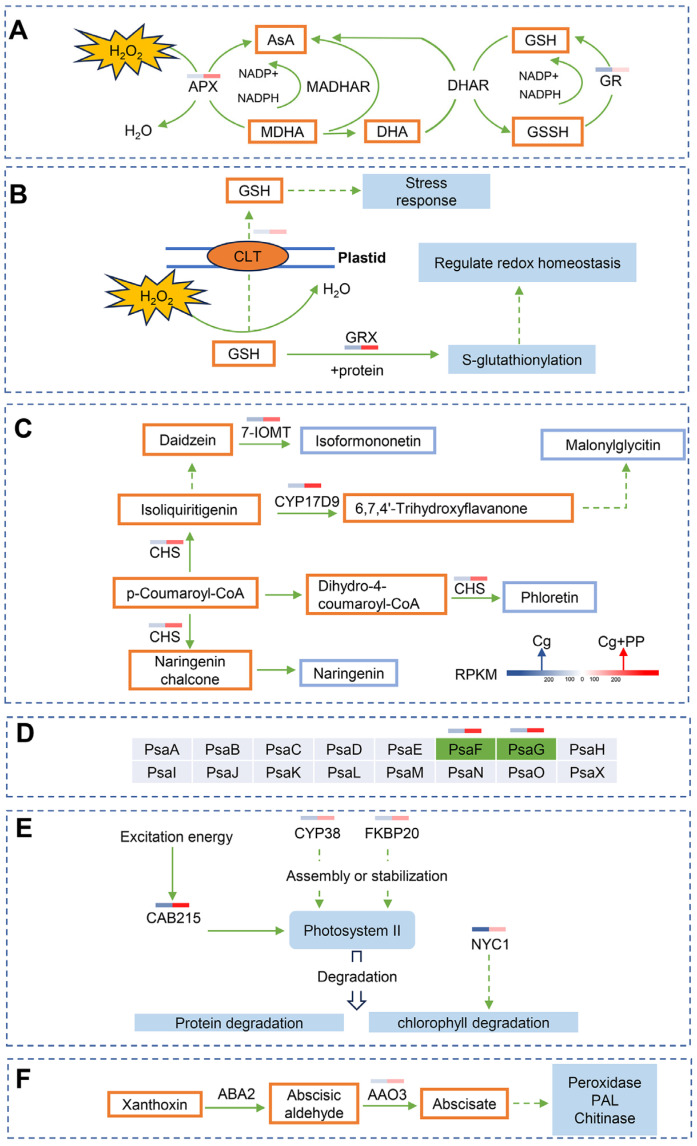
Relative expression of DEGs in the pathways involved in production of antioxidant and chloroplast organization of stylo treated by phloretin and pterostilbene during *C. gloeosporioides* infection. The mean FPKM (fragments per kilobase of transcript per million mapped reads) values for the DEGs were calculated from three biological replicates. The progression of the color scale from blue to red represents an increase in the FPKM values. (**A**) AsA-GSH cycles. APX: ascorbate peroxidase, AsA: ascorbate acid, MDHA: monodehydroascorbate, MADHAR: monodehydroascorbate reductase, DHA: Dehydroascorbic acid, DHAR: dehydroascorbate reductase, GSH: glutathione, GSSH: oxidized glutathione, GR: glutathione reductase. (**B**) GSH-related pathways. CLT: CRT-like transporter, GRX: Glutaredoxin. (**C**) Flavonoids biosynthesis. 7-IOMT: isoflavone-7-O-methyltransferase, CYP17D9: flavonoid 6-hydroxylase, CHS: chalcone synthase. (**D**) Subunits of photosystem I. (**E**) Photosystem II-related. CYP38: cyclophilin38, FKBP20: peptidyl-prolyl cis-trans isomerase FKBP20-2, CAB215: chlorophyll a-b binding protein 215, NYC1: chlorophyll b reductase NYC1. (**F**) Carotenoid metabolic pathway. ABA2: short-chain alcohol dehydrogenase 2, AAO3: aldehyde oxidase 3, ABA: abscisic acid, PAL: phenylalanine ammonia lyase.

**Figure 5 ijms-25-02701-f005:**
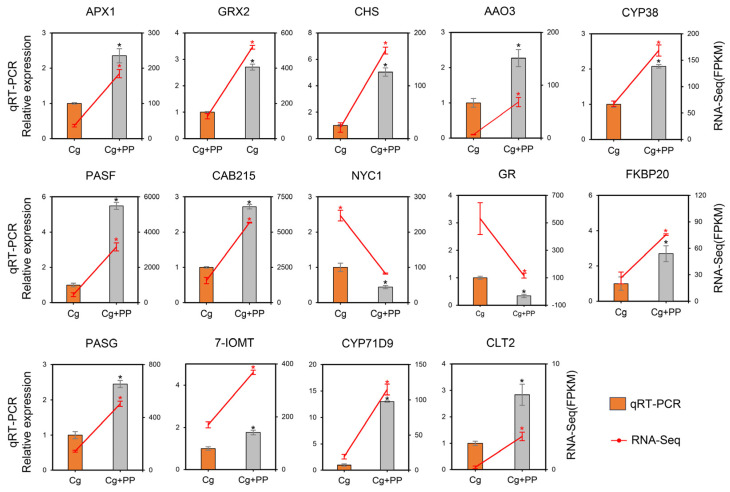
Validation of DEG expression by qRT-PCR. Expression levels of DEGs were normalized based on transcript levels of *UBCE1* gene. Columns represent the qRT-PCR results, and the line chart represents the RNA-Seq data. The asterisk “*” represents significant difference between Cg and Cg + PP according to Student’s *t*-test at *p* < 0.05. Data were pooled from three biological replicates.

**Figure 6 ijms-25-02701-f006:**
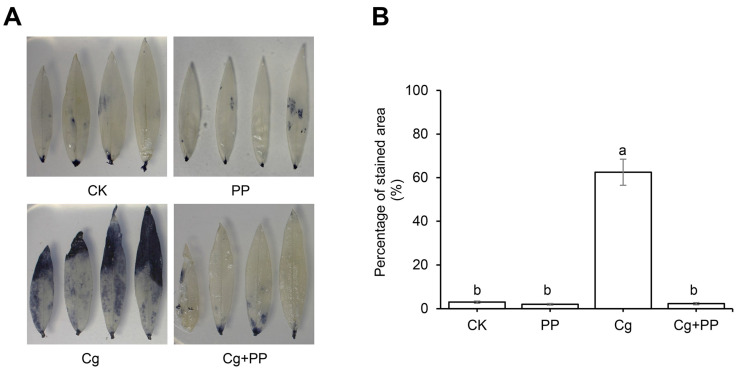
The accumulation of ROS in the leaves of stylo under different treatments. (**A**) NBT staining of leaves. (**B**) Statistics of percentage of staining area. The leaf samples were collected at 72 hpi. CK, uninoculated samples; PP, uninoculated samples treated with phloretin and pterostilbene; Cg, samples inoculated with *C. gloeosporioides*; Cg + PP, samples inoculated with *C. gloeosporioides* together treated with phloretin and pterostilbene. Data were the mean ± SD of three biological replicates. Values with different letters are significantly different according to ANOVA (*p* < 0.05).

**Figure 7 ijms-25-02701-f007:**
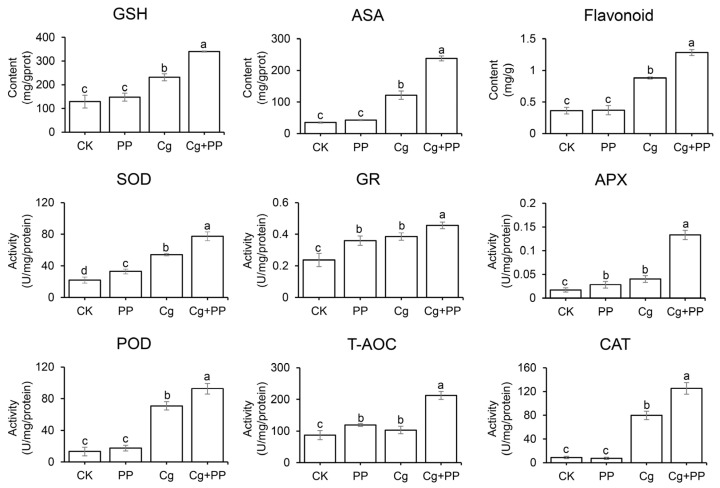
Detection of the content of antioxidant substances and the activities of antioxidant enzymes of stylo under different treatments. Data were the mean ± SD of three biological replicates. Values with different letters are significantly different according to ANOVA (*p* < 0.05).

**Figure 8 ijms-25-02701-f008:**
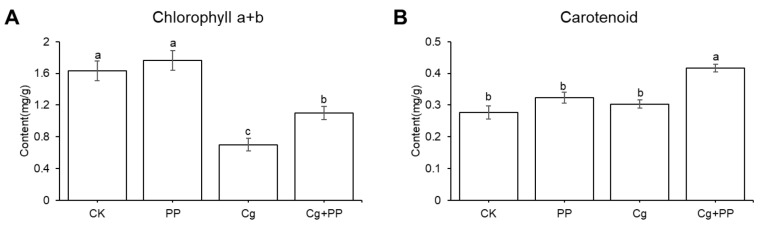
Determination of pigment contents under different treatments. The leaf samples were collected from 72 hpi after inoculation with *C. gloeosporioides*. Data were mean ± SD of sample leaves from three biological replications. Different letters indicate significant difference at the *p* < 0.05 level based on the ANOVA.

## Data Availability

The original data can be found in NCBI (PRJNA1058883).

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
