# Peer review of "Transcriptomic and Physiological Analysis of the Effects of Exogenous Phloretin and Pterostilbene on Resistance Responses of Stylosanthes against Anthracnose"

_ijms, 2024, doi:10.3390/ijms25052701_

Round 1
Reviewer 1 Report
Comments and Suggestions for Authors
The study on anthracnose caused by C. gloeosporioides and the application of the combination treatment of phloretin and pterostilbene (PP) presents promising results for controlling the disease. The in vitro effectiveness of PP in inhibiting conidial germination and mycelial growth of C. gloeosporioides is now supported by in vivo evidence. The positive impact of exogenous PP treatment on limiting the growth of C. gloeosporioides and mitigating anthracnose damage on stylo is a significant finding.
The article is well-designed, written, and presented. I would recommend adding more detail to the objective, and the discussion should be more connected not fragmented.
Reviewer 2 Report
Comments and Suggestions for Authors
Dear Authors,
I believe that the work is well written but requires minor improvement.
1) In the Introduction, I suggest presenting Stylosanthes spp. in more detail (in a few sentences).
2) Latin names should be written in italics, e.g. L 59-60.
3) I suggest rewording the text in lines 75-79. The research hypothesis and purpose of the research conducted should be indicated here, rather than the results achieved.
4) Units of measurement should be given according to the SI system, as required by the IJMS.
5) In subsection 4.9 Statistical analysis, the Authors indicated the statistical programs used for calculations (Microsoft Excel and SPSS 434), but did not indicate the statistical analysis methods used to process the data.
Reviewer 3 Report
Comments and Suggestions for Authors
Well written paper , only minor English editing is necessary. Data presented are new and interesting. The experimental design is correct, results from RNAseq are supported by PCR analysis of selected DEGs, histochemical visualization of ROS and biochemical analysis of ROS scavenging enzymes and compounds, as well as leaf pigment content. Results are properly discussed without any speculations.
Introduction – straight to the point, concise and logical. Minor remarks:
Why combination of two phenolic compounds with a probably similar action, is the effect synergistic or additive? Separately the two compounds were tested with different sets of pathogens. The preliminary study on the “Antifungal activity of 13 phenylpropanoid metabolites against six Colletotrichum species” is not available on the web in order to consult it. Previous research on separate or combined treatment should be briefly described in the Introduction. I also wonder if the observed is rather a delay in the infection or real mitigation of the symptoms in a long run.
Instead of aim of the study the results are stated in brief. The aim is pointed out at the beginning of the Discussion (the first paragraph).
Refs – Introduction – 26 refs. Ref 27 is in MMs which are at the end – should be rearranged.
MMs 4.1. Stylo material and Colletotrichum gloeosporioides inoculation – The growth of plants should be described in more details (not just – as previously described)
Results - Refs 33,34,35 – should not be in Results but could be placed in the Discussion.
Comments on the Quality of English LanguageMinor English editing is necessary (plural for example).
